# Effect of Hesperidin on Barrier Function and Reactive Oxygen Species Production in an Oral Epithelial Cell Model, and on Secretion of Macrophage-Derived Inflammatory Mediators during *Porphyromonas gingivalis* Infection

**DOI:** 10.3390/ijms241210389

**Published:** 2023-06-20

**Authors:** Patricia Milagros Maquera-Huacho, Denise Palomari Spolidorio, John Manthey, Daniel Grenier

**Affiliations:** 1Oral Ecology Research Group, Faculty of Dentistry, Université Laval, Quebec City, QC G1V 0A6, Canada; daniel.grenier@greb.ulaval.ca; 2Department of Physiology and Pathology, School of Dentistry, São Paulo State University (Unesp), Araraquara 14801-903, SP, Brazil; denise.mp.spolidorio@unesp.br; 3School of Medicine, Faculty of Health Sciences, National University of Moquegua, Moquegua 18001, Peru; 4U.S. Horticultural Research Laboratory, Agricultural Research Service, USDA, Fort Pierce, FL 34945, USA; jmanthey56@gmail.com

**Keywords:** hesperidin, inflammation, oxidative stress, macrophage, keratinocyte, *Porphyromonas gingivalis*

## Abstract

*Porphyromonas gingivalis* is a periodontopathogenic bacterium that can adhere to and colonize periodontal tissues, leading to an inflammatory process, and, consequently, tissue destruction. New therapies using flavonoids, such as hesperidin, are being studied, and their promising properties have been highlighted. The aim of this study was to evaluate the effect of hesperidin on the epithelial barrier function, reactive oxygen species (ROS) production, and on the inflammatory response caused by *P. gingivalis* in in vitro models. The integrity of the epithelial tight junctions challenged by *P. gingivalis* was determined by monitoring the transepithelial electrical resistance (TER). *P. gingivalis* adherence to a gingival keratinocyte monolayer and a basement membrane model were evaluated by a fluorescence assay. A fluorometric assay was used to determine the ROS production in gingival keratinocytes. The level of pro-inflammatory cytokines and matrix metalloproteinases (MMPs) secretion was evaluated by ELISA; to assess NF-κB activation, the U937-3xjB-LUC monocyte cell line transfected with a luciferase reporter gene was used. Hesperidin protected against gingival epithelial barrier dysfunction caused by *P. gingivalis* and reduced the adherence of *P. gingivalis* to the basement membrane model. Hesperidin dose-dependently inhibited *P. gingivalis*-mediated ROS production by oral epithelial cells as well as the secretion of IL-1β, TNF-α, IL-8, MMP-2, and MMP-9 by macrophages challenged with *P. gingivalis*. Additionally, it was able to attenuate NF-κB activation in macrophages stimulated with *P. gingivalis*. These findings suggest that hesperidin has a protective effect on the epithelial barrier function, in addition to reducing ROS production and attenuating the inflammatory response associated with periodontal disease.

## 1. Introduction

A large diversity of microorganisms reside in the oral cavity and colonize the teeth, tongue, cheeks, gingival sulcus, tonsils, hard palate, and soft palate [1]. In periodontal disease, an alteration in the microbial composition and/or pathogenicity occurs, resulting in microbial dysbiosis [2]. Indeed, this condition can induce an exacerbated inflammatory response associated with the secretion of cytokines and matrix metalloproteinases (MMPs) [3]. As the disease progresses, the destruction of the tooth-supporting structures is promoted, including the periodontal ligament and the alveolar bone [3,4]. Periodontal tissue damage is also caused by an excessive production of reactive oxygen species (ROS), which contribute to cellular apoptosis and cause oxidative stresses [5]. As described by Socransky et al. [6], “the red complex,” composed of *Porphyromonas gingivalis*, *Tannerella forsythia*, and *Treponema denticola*, has been strongly associated with the severity of periodontitis. Among these periodontopathogenic bacteria, the gram-negative bacterium *P. gingivalis* is thought to cause the dysbiosis condition and to initiate the inflammatory process of periodontal disease [7].

The oral epithelium is the main structure of the oral cavity that provides physical, chemical, and immunological barriers against chemical and microbial challenges [8]. During periodontal disease, colonization of the gingival sulcus by periodontopathogenic bacteria leads to the release of microbial virulence factors and toxins, including lipopolysaccharide (LPS), that stimulate the inflammatory process and tissue destruction [9]. *P. gingivalis*, known as a key periodontopathogenic bacterium, can express adhesive components [10] and damage the epithelial barrier through proteolytic degradation of the junction proteins [11]. Gingipains, including Arg-gingipains A and B and Lys-gingipain, are the major extracellular and cell-bound proteolytic activities produced by *P. gingivalis* [12]. More specifically, gingipains mediate collagen degradation, mainly type I collagen, which is particularly abundant in periodontal tissues [12]. In addition, gingipains induce host defense perturbation through the degradation of cellular signaling molecules and the inactivation of cellular functions [12,13].

To enhance the outcome of periodontal disease treatments, several adjuncts to non-surgical periodontal therapy have been proposed, including local delivery drugs, such as antibiotics and antimicrobial agents. In recent years, natural products have been extensively studied due to their high safety, cost-effectiveness, and beneficial properties, such as their anti-inflammatory, antioxidant, and antimicrobial properties. In this context, hesperidin is a flavonoid glycoside found in large amounts in citrus fruits, such as sweet oranges, lemons, limes, and tangerines [14]. Hesperidin has been reported to exert beneficial effects on various diseases, including neurological, cardiovascular, and other disorders, due to its anti-inflammatory, anti-oxidative, and anti-cancer properties [15]. Concerning periodontal disease, in vivo studies using rodent models have shown that hesperidin can promote an ameliorative effect on alveolar bone loss and attenuate the intensity of the inflammatory process [16,17]. 

Despite evidence suggesting a promising effect of hesperidin for periodontal disease, additional studies are necessary to elucidate the underlying mechanisms of action of this flavonoid. Therefore, the purpose of this study was to investigate the effects of hesperidin on *P. gingivalis*-mediated barrier function damage and ROS production in an oral epithelial model. Moreover, the ability of hesperidin to attenuate the expression of inflammatory cytokines and MMPs by *P. gingivalis*-stimulated macrophages was studied.

## 2. Results

### 2.1. Effect of Hesperidin on the Disruption of Epithelial Barrier Integrity

The epithelial barrier is a complex and critical structure that protects against tissue invasion by periodontopathogenic bacteria. In this context, we evaluated the ability of hesperidin to protect the integrity of the oral epithelial barrier without stimulus or against *P. gingivalis* by monitoring TER values over a 24 h period. The TER measurement is a simple and noninvasive method used to monitor and evaluate the epithelial barrier’s integrity. Initially, we performed an MTT assay to determine non-cytotoxic concentrations of hesperidin. As shown in Figure 1, concentrations ≤ 1280 µM of hesperidin did not significantly decrease cell viability after 48 h of exposition.

The statistical analysis showed that hesperidin was able to increase TER values following a 2, 4, and 8 h exposure compared with the control group (Figure 2A). In detail, hesperidin at 640 µM showed an increase in the TER by 80.61% and 28.31% after 2 and 4 h, respectively. Furthermore, high TER values were observed after treatment with hesperidin at 1280 µM, resulting in an increase of 114.38%, 41.54%, and 16.11% after 2, 4, and 8 h. In general, hesperidin showed a dose-dependent effect. Treating the oral epithelial barrier model simultaneously with hesperidin and *P. gingivalis* was associated with significantly higher TER values after 8 and 24 h treatments compared to treating the model with *P. gingivalis* alone (Figure 2B). More specifically, after an 8 h treatment, hesperidin at 640 µM showed an increase in the TER by 15.36%, while hesperidin at 1280 µM resulted in an increase of 79.81% after 8 h. Extending the incubation period to 24 h showed an increase in the TER of 13.22% and 61.13% for hesperidin at 640 µM and 1280 µM, respectively. Treating the oral epithelial barrier model for 24 h with *P. gingivalis* in the absence of hesperidin showed a significant decrease in TER values (187.90%) compared to the control group (without *P. gingivalis* stimulus). An increase in TER values is observed after 2, 4, and 8 h of exposure with *P. gingivalis* compared to the control group. On the other hand, the TER values reached their maximum confluency after a 24 h treatment of the control group cells, suggesting the integrity of the cell layer.

### 2.2. Effect of Hesperidin on the Adherence of P. gingivalis to Human Oral Epithelial Cells and a Basement Membrane Model

We determined the effect of hesperidin on the capacity of *P. gingivalis* to adhere to the human oral epithelial cells and the basement membrane. We used the FITC labeling of microorganisms, which is a common technique used extensively to evaluate the interaction between bacteria and cells. Briefly, FITC covalently binds amino acids present on the N terminus of proteins and lysine residues. Figure 3 reports the effect of hesperidin on the adherence of *P. gingivalis* to human oral epithelial cells (A) and a basement membrane model (B). On the one hand, hesperidin did not show any capacity to reduce the adherence of *P. gingivalis* to human oral epithelial cells (Figure 3A). On the other hand, the flavonoid dose-dependently inhibited the adherence of *P. gingivalis* to the basement membrane model (Figure 3B). Hesperidin at 640 µM and 1280 µM reduced bacterial adhesion by 21.11% and 43.06%, respectively. 

### 2.3. Effect of Hesperidin on ROS Production

The excessive production of ROS by human oral epithelial cells may contribute to the destruction of the tooth-supporting structures [18]. As reported in Figure 4, the exposure of oral epithelial cells to *P. gingivalis* (MOI = 1000) time-dependently mediated ROS production. More specifically, a 3-fold increase was observed following a 5 h contact time. The presence of hesperidin (320, 640, and 1280 µM) reduced *P. gingivalis*-induced ROS production between 11.0% and 15.9% (5 h contact). 

The effect of hesperidin on ROS production by human oral epithelial cells co-stimulated with *P. gingivalis* and H_2_O_2_ was also evaluated. Figure 5 shows high levels of ROS production by oral epithelial cells treated with *P. gingivalis* and H_2_O_2_; this was a 102.6-fold increase compared to the control group after a 5 h contact time. Epithelial cells stimulated with *P. gingivalis* and H_2_O_2_ separately showed only an 11.4- and 3.3-fold increase in the production of ROS compared to the control group, thus suggesting a synergistic effect between *P. gingivalis* and H_2_O_2_. However, the presence of hesperidin did not significantly attenuate ROS production in this synergistic model. 

### 2.4. Effect of Hesperidin on Cytokine and MMP Expression

To assess the effect of hesperidin on the inflammatory response associated with periodontitis, macrophages were pre-treated for 2 h with non-cytotoxic concentrations of hesperidin and stimulated with *P. gingivalis* (MOI of 100) for 24 h. First, hesperidin concentrations of 20, 40, and 80 µM were determined as being non-cytotoxic for macrophages in an MTT colorimetric assay (Figure 6). 

Subsequent experiments using ELISA assay were performed to assess the levels of selected cytokines and MMPs secreted by macrophages. As shown in Figure 7, increased secretion of IL-6, IL-8, TNF-α, and IL-1β by macrophages was observed when exposed to *P. gingivalis*, compared with unstimulated controls. An increase of 17.6-, 8.5-, 138.5-, and 4.2-fold was found for IL-6, IL-8, TNF-α, and IL-1β, respectively. Hesperidin dose-dependently down-regulated the secretion of IL-8, TNF-α, and IL-1β. However, this effect was not observed for IL-6 secretion. More specifically, hesperidin at a concentration of 80 µM was able to inhibit 29.9, 42.0, and 30.2% of IL-8, TNF-α, and IL-1β secretion, respectively.

The effect of hesperidin on MMP secretion is shown in Figure 8. The results indicate that levels of MMP-2, MMP-8, and MMP-9 increased by 2.4-, 1.6-, and 1.8-fold after *P. gingivalis* exposition when compared to unstimulated controls. Hesperidin at a concentration of 80 µM in *P. gingivalis*-stimulated macrophages decreased the secretion of MMP-2 and MMP-9 by 74.4 and 30.4%, respectively. This effect was also dose-dependent. However, hesperidin did not show any significant inhibition of MMP-8 secretion by macrophages stimulated with *P. gingivalis*.

### 2.5. Effect of Hesperidin on NF-κB Activation

Lastly, to investigate the mechanism responsible for the inhibitory effect of hesperidin on the inflammatory response of *P. gingivalis*-stimulated macrophages, we verified NF-κB activation after treatment with hesperidin. To this end, an alternative model using the U937-3xκB-LUC monocytic cell line was used. Figure 9 shows a significant inhibition (13.4%) of NF-κB activation after exposure to hesperidin at a concentration of 80 µM.

## 3. Discussion

In periodontal disease, damage and destruction of periodontal tissues are caused mainly by an inappropriate host response to microorganisms and their products. *P. gingivalis* can disturb the epithelial integrity and invade the deeper periodontal tissues, triggering an inflammatory response [19]. In this context, therapies involving agents that enhance the oral mucosa barrier and that exhibit anti-inflammatory and antioxidant properties are promising for periodontal disease treatment. Among them, hesperidin is an attractive flavonoid with excellent biological properties that can modulate the epithelial cellular immune response [15]. As such, the present study aimed to investigate the effect of hesperidin on *P. gingivalis*-mediated damage to the gingival epithelial barrier function, ROS production, and inflammatory response using in vitro models.

The gingival epithelium is an important barrier against bacterial invasion, and for this reason, it becomes an important early line of defense in the oral cavity [20]. Considering this concept, diverse flavonoids with the capacity to enhance the oral epithelial barrier function are interesting candidates for the treatment or prevention of periodontal disease. In the present in vitro study, we reported interesting results of the TER after treatment with hesperidin. More specifically, hesperidin showed a significant increase in the TER following a short exposure (2, 4, and 8 h). This is likely due to the fact that hesperidin may improve and augment the formation of tissue junctions. Additional studies are required to confirm this hypothesis. Although the capacity of hesperidin to enhance or protect tissue barrier function has not been investigated, Guo et al. [21] showed the capacity of hesperidin to enhance barrier integrity as well as epithelial permeability in Caco-2 cells.

The ability of *P. gingivalis* to invade epithelial cells is triggered via the binding of fimbriae to cellular adhesion molecules, thereby disrupting the epithelial integrity [13,22]. These cell–cell interactions are critical for the innate immune response against microbial and toxic challenges [23]. In the present study, using an in vitro epithelial barrier model stimulated with *P. gingivalis*, cells revealed a progressive increase of the TER after 2 and 8 h of incubation. This increase in the TER after exposure to *P. gingivalis* may be due to the increase of bacterial adhesion to the oral epithelial cells, leading to an additional resistance to the TER. After 24 h, the TER values began to decrease, indicating damage in the function of the epithelial barrier. These data are in agreement with a previous in vitro study that showed the same tendency in TER values [19]. On the other hand, our results showed that cells infected with *P. gingivalis* and treated with hesperidin showed a positive effect after 24 h by increasing the TER values when compared to the cells infected with bacteria alone. It remains unclear what is the exact mechanism of the protective effect of hesperidin on the epithelial barrier integrity in our model. 

*P. gingivalis* can attach to the periodontal tissues, and several investigations have been reported regarding its adhesion to epithelial cells [24]. Using an in vitro study, we showed that *P. gingivalis* was able to adhere to oral epithelial cells. However, hesperidin did not show a capacity to decrease this adhesion. This effect may be due to the fact that the proteolytic activity may easily deteriorate the epithelial cells. However, other tests are required to confirm this hypothesis. In contrast, using a basement membrane model, hesperidin was able to decrease the adherence of *P. gingivalis*. As such, our data suggest that hesperidin may alter the ability of *P. gingivalis* to adhere to several extracellular matrix proteins, including laminin and type IV collagen, which are constituents of the basement membrane model. The manufacturer indicates that the membrane provides greater protection against invasion by other cells. Therefore, we hypothesize that the treatment of the basement membrane with hesperidin increased its protective effect against the invasion of *P. gingivalis*, which explains our results. To the best of our knowledge, no study has previously evaluated the effect of hesperidin on the adherence of periodontopathogens to epithelial cells or periodontal tissues/cells.

During periodontal disease, the exacerbated production of ROS is responsible for cell apoptosis and dysfunction, alveolar bone loss, and periodontal inflammation [25]. In this way, the attenuation or inhibition of ROS production is a relevant and important event to regulate several signaling pathways, and, consequently, the inflammatory response [26]. In the present study, we proposed two ROS production models using *P. gingivalis* alone and *P. gingivalis* + H_2_O_2_ (exacerbated ROS production model) as the stimulus. H_2_O_2_ is known as a major component of ROS and is widely used as an inducer of oxidative stress. Based on our results, hesperidin can inhibit ROS production in response to *P. gingivalis*. However, hesperidin did not show down-regulating effects in an event with increased levels of ROS production. Previous in vitro studies have shown that hesperidin treatment was able to reduce the by-products of lipid peroxidation in the human erythrocyte membrane [27,28]. Moreover, hesperidin attenuates the production of intracellular ROS in macrophages stimulated with lipopolysaccharide [29]. On the other hand, in vivo studies using models of supplementation in rats have shown that hesperidin reduces the expression of levels of ROS and thiobarbituric acid reactive substances (TBARS) and increases the activity of antioxidants [30,31,32]. Additionally, Lim et al. [33] demonstrated the marked ability of hesperidin to inhibit high-glucose-induced intracellular ROS production in SH−SY5Y neuronal cells. Although the literature shows its promising antioxidant properties, our results using *P. gingivalis* bacteria associated or not with H_2_O_2_ are not in accordance. This may be because *P. gingivalis* can invade and survive in epithelial cells and then accumulate a hemin layer on the cell surface, which has the function of providing oxidative stress protection [34]. Therefore, more studies are necessary to understand the antioxidant mechanism and effect of hesperidin in ROS production.

Because *P. gingivalis* plays an important role in the induction of inflammatory processes, we evaluated the anti-inflammatory activity of hesperidin against this periodontopathogen. Using a monocyte–macrophage model stimulated with bacteria, we found evidence that hesperidin was able to inhibit the secretion of IL-1β, TNF-α, and IL-8. Previously, the anti-inflammatory potential of hesperidin has been demonstrated using various cell models [35,36], and the data are in accordance with our results. In contrast to our results, Li et al. [37], among the cytokines evaluated, also showed an inhibition of IL-6 after treatment with hesperidin in an in vivo protocol.

High levels of MMPs, such as MMP-1, -2, -8, -9, and -13, are related to the severity and tissue destruction of periodontal disease [38,39]. In this context, the down-regulation of MMP-2 and -9 by hesperidin, as observed in our results, may promote an anti-collagenase effect. These findings are in agreement with previous studies that reported the effect of hesperidin in inhibiting the expression of these MMPs [40,41,42]. Additionally, we showed for the first time that hesperidin was not able to reduce the expression of the MMP-8 level. There are no previous data exploring the mechanism of hesperidin on MMP-8 expression. Overall, our results showed selective inhibition of IL-1β, TNF-α, IL-8, MMP-2, and MMP-9 associated with hesperidin, which could represent a promising novel approach to the treatment or prevention of periodontal disease.

The NF-κB pathway is a key component of the inflammatory response that can be triggered by bacteria and their toxic products, resulting in cytokine expression [43]. Therefore, the capacity of compounds to inactivate this pathway becomes an important characteristic for the modulation of the inflammatory response. In this context, our results showed that hesperidin was able to inhibit NF-κB activation in the human monoblastic leukemia cell line transfected with a luciferase gene coupled to a promoter of three NF-κ B binding sites. As expected, these findings suggest that the inhibitory effect of hesperidin on pro-inflammatory cytokine expression may be mediated through the inhibition of the NF-κB pathway. Similarly, Sato et al. [44] suggested that hesperidin has a significant effect on the regulation of the NF-κB signaling pathway.

## 4. Materials and Methods

### 4.1. Hesperidin

Hesperidin was obtained from the USDA-ARS Horticultural Research Laboratory (Fort Pierce, FL, USA). A stock solution at a concentration of 1 M was prepared in dimethylsulfoxide (DMSO) and stored at 4 °C in the dark. In all assays described below, the final concentration of DMSO in the culture medium was ≤0.1% (*w*/*v*).

### 4.2. Bacteria and Growth Conditions

The reference strain *P. gingivalis* ATCC 33277 was used in this study. Bacteria were grown in an anaerobic chamber (80% N_2_, 10% CO_2_, 10% H_2_) at 37 °C in Todd-Hewitt broth (THB; Becton Dickinson and Company, Sparks, MD, USA) containing 0.001% (*w*/*v*) hemin and 0.0001% (*w*/*v*) vitamin K.

### 4.3. Cell Cultures

The B11 human oral epithelial cell line [45], which was kindly provided by S. Gröger (Justus Liebig University Giessen, Germany), was cultivated in keratinocyte serum-free medium (K-SFM; Life Technologies Inc., Burlington, ON, Canada) supplemented with 50 µg/mL of bovine pituitary extract, 5 ng/mL of human epidermal growth factor, 100 µg/mL of penicillin G/streptomycin, and 0.5 µg/mL of amphotericin B. The U937 human monocyte cell line (CRL-1593.2), which was purchased from the American Type Culture Collection (Manassas, VA, USA), was cultivated in Roswell Park Memorial Institute 1640 medium (RPMI; Life Technologies Inc.) supplemented with 10% heat-inactivated fetal bovine serum (FBS) and 100 µg/mL of penicillin G/streptomycin. The human monoblastic leukemia cell line U937-3xκB-LUC, a subclone of the U937 cell line transfected with a luciferase gene coupled to a promoter of three NF-κB binding sites [46], was kindly provided by R. Blomhoff (University of Oslo, Norway). This cell line was cultured in RPMI-1640 supplemented with 10% FBS, 100 μg/mL of penicillin G/streptomycin, and 75 μg/mL of hygromycin B. All cell cultures were incubated in a humidified incubator with a 5% CO_2_ atmosphere at 37 °C. 

### 4.4. Oral Epithelial Barrier Integrity 

To investigate the effect of hesperidin on the epithelial barrier integrity challenged or not with *P. gingivalis*, the B11 human oral epithelial cell line [45] was used. First, non-cytotoxic concentrations of hesperidin were determined using MTT (3-[4,5-diethylthiazol-2-yl]-2,5diphenyltetrazolium bromide) colorimetric assay (Roche Diagnostics, Laval, QC, Canada) according to the manufacturer’s instructions following a 48 h treatment. The epithelial cells were seeded onto a Transwell^TM^ clear polyester membrane insert (6.5 mm diameter, 0.4 μm pore size; Corning Co., Cambridge, MA, USA) (3 × 10^5^ cells per insert). Basolateral and apical compartments were filled with 0.6 mL and 0.1 mL of culture medium, respectively. Then, the plates were incubated at 37 °C in a 5% CO_2_ atmosphere for 72 h. After this period, the culture medium was replaced with fresh antibiotic-free K-SFM, and the plates were further incubated for 16 h. Then, in the first setup, non-cytotoxic concentrations of hesperidin (640 and 1280 µM) were added to the apical compartments, and in the second setup, *P. gingivalis* at a multiplicity of infection (MOI) of 10^4^ prepared in antibiotic-free K-SFM was associated or not with non-cytotoxic concentrations and added to the apical compartments. Finally, the integrity of the epithelial tight junctions was determined by monitoring the transepithelial electrical resistance (TER) using an ohmmeter (EVOM2, World Precision Instruments, Sarasota, FL, USA). TER was determined after 0, 2, 4, 8, and 24 h of incubation at 37 °C in a 5% CO_2_ atmosphere. Resistance values were calculated in Ohms (Ω)/cm^2^ by multiplying the resistance values by the surface area of the membrane filter. Results were expressed as a percentage of the basal control value measured at time 0 (100% value). Assays were carried out in triplicate in two independent experiments, and the means ± standard deviations were calculated.

### 4.5. Adherence of P. gingivalis to Human Oral Epithelial Cells and a Basement Membrane Model

The effect of hesperidin on the adherence of *P. gingivalis* to human oral epithelial cells and a basement membrane model (Cultrex Basement Membrane Extract [BME]; R&D Systems, Minneapolis, MN, USA) was evaluated, as previously described [47]. First, a 24 h culture of *P. gingivalis* was harvested by centrifugation (9000× *g* for 5 min), washed in 10 mm phosphate-buffered saline (PBS; pH 7.2), and suspended in K-SFM medium or 50 mM sodium bicarbonate buffer (pH 8) containing 0.03 mg/mL of fluorescein isothiocyanate (FITC) to assess adherence to the B11 human oral epithelial cell or the basement membrane model, respectively. Then, the *P. gingivalis* suspension was incubated in the dark at 37 °C for 30 min with constant shaking. Thereafter, the labeled bacteria were washed three times by centrifugation (9000× *g* for 5 min). In the first analysis, oral epithelial cells (1 × 10^6^ cells/mL) were seeded in sterile black wall, clear flat bottom 96-well microplates (Greiner Bio-One North America, Monroe, NC, USA) and incubated overnight at 37 °C in a 5% CO_2_ atmosphere to allow cell adherence. Then, the medium was removed, and the cell monolayers were pre-incubated with hesperidin (320, 640, or 1280 µM; in K-SFM) for 30 min before adding FITC-labeled *P. gingivalis* at an MOI of 1000. The cells were incubated at 37 °C in a 5% CO_2_ atmosphere. After 4 h, unbound bacteria were removed, the wells were washed with PBS, and the number of bacteria that adhered was determined by relative fluorescence units (RFU) using a Synergy 2 microplate reader (BioTek Instruments, Winooski, VT, USA) (excitation wavelength of 495 nm and an emission wavelength of 525 nm). In a second analysis, the BME containing laminin, collagen IV, entactin, and heparin sulfate proteoglycan was diluted 1:10 in ice-cold PBS, and 100 μL was added to the wells of 96-well clear bottom black microplates. The microplates were maintained for 2 h at room temperature to allow gelification. Then, the wells were washed with PBS, and two-fold serial dilutions of hesperidin (320, 640, or 1280 µM in PBS) were added for 30 min. Thereafter, FITC-labeled *P. gingivalis* (OD660 = 0.5) was added to each well (100 μL), and the microplate was maintained at 37 °C for 4 h. The number of bacteria that adhered to the basement membrane model was determined as described above. Wells without *P. gingivalis* were used as controls to measure basal auto-fluorescence and wells without hesperidin were used to determine 100% adherence values. Adherence assays were carried out in triplicate in three independent experiments, and the means ± standard deviations were calculated.

### 4.6. Reactive Oxygen Species Production by Human Oral Epithelial Cells

The B11 human oral epithelial cell line (3 × 10^5^ cells/well) was seeded in black wall, clear flat bottom 96-well microplates (Greiner Bio-One North America, Monroe, NC, USA) and incubated overnight at 37 °C in a 5% CO_2_ atmosphere. A fluorometric assay to monitor the oxidation of 2′,7′-dichlorofluorescein-diacetate (DCF-DA; Sigma-Aldrich Canada Co., Oakville, ON, Canada) into a fluorescent compound was used to measure ROS production. A 40 mM stock solution of DCF-DA was freshly prepared in DMSO. The cells were washed with Hank’s balanced salt solution (HBSS; HyClone Laboratories, Logan, UT, USA) and incubated for a further 30 min in the presence of 100 µM DCF-DA in HBSS. Keratinocytes were washed with HBSS to remove the excess DCF-DA. Then, the cells were treated with *P. gingivalis* at an MOI of 100 either in the presence or absence of 1 mM hydrogen peroxide (H_2_O_2_) and in the absence or presence of hesperidin (320, 640, or 1280 µM) previously prepared in HBSS. ROS production by monitoring fluorescence emission was recorded every 20 min for 5 h using a Synergy 2 microplate reader (BioTek Instruments) (a 485 nm excitation filter and a 528 nm emission filter). The assays were carried out in triplicate in three independent experiments, and the means ± standard deviations were calculated. 

### 4.7. Cytokine and Matrix Metalloproteinases (MMPs) Secretion by Macrophages

To evaluate the effect of hesperidin on the secretion of pro-inflammatory cytokines (IL-1β, IL-6, IL-8, TNF-α) and MMPs (MMP-2, MMP-8, MMP-9), U937 human monocytes (CRL-1593.2; American Type Culture Collection, Manassas, VA, USA) were used. Non-cytotoxic concentrations of hesperidin were first determined using MTT (3-[4,5-diethylthiazol-2-yl]-2,5diphenyltetrazolium bromide) colorimetric assay (Roche Diagnostics, Laval, QC, Canada) according to the manufacturer’s instructions following a 24 h treatment. The monocytes (1 × 10^6^ cells/mL) were differentiated into adherent macrophage-like cells by incubation in RPMI-10% FBS containing 100 ng/mL of phorbol myristic acid (PMA; Sigma-Aldrich Canada Co.) for 48 h into 12-well microplates. The adherent macrophage-like cells were washed to remove the remaining PMA and non-adherent cells and were maintained in RPMI-10% FBS without PMA for 24 h. Thereafter, the medium was changed and the cells were maintained in RPMI-1% FBS and incubated overnight at 37 °C in a 5% CO_2_ atmosphere. The macrophage-like cells were pre-treated for 2 h with hesperidin (20, 40, or 80 µM) previously prepared in RPMI-1% FBS, and they were stimulated with *P. gingivalis* cells at an MOI of 100 for 24 h. Then, the culture medium supernatants were collected and kept at −20 °C until use. Culture medium collected from cells without treatment with hesperidin and bacteria was used as the control. Enzyme-linked immunosorbent assay (ELISA) kits (R&D Systems, Minneapolis, MN, USA; Invitrogen, Thermo Fisher Scientific Inc., Waltham, MA, USA) were used to determine cytokine and MMP concentrations according to the manufacturer’s protocols. Assays were performed in triplicate in three independent experiments, and the means ± standard deviations were calculated. 

### 4.8. Activation of NF-κB in P. gingivalis-Stimulated Monocytes 

To evaluate the effect of hesperidin on *P. gingivalis*-induced NF-κB activation, the human monoblastic leukemia cell line U937-3xκB-LUC was used. The monocytes were seeded (10^5^ cells/well) into wells of the black wall, black bottom, 96-well microplates (Greiner Bio-One North America) and pre-incubated with hesperidin (20, 40, or 80 µM) for 30 min. Thereafter, the monocytes were stimulated with *P. gingivalis* at an MOI of 100 for 6 h. Wells containing monocytes without *P. gingivalis* or hesperidin were used as controls. Bright-Glo reagent (Promega Corporation, Durham, NC, USA) was used according to the manufacturer’s protocol to measure luciferase activity and to determine NF-κB activation. Luminescence was monitored using a Synergy 2 microplate reader (BioTek Instruments). Assays were performed in triplicate in three independent experiments, and the means ± standard deviations were calculated. 

### 4.9. Statistical Analysis

A one-way ANOVA with a post hoc Bonferroni multiple comparison test was used to analyze the data. Results were considered statistically significant at *p* < 0.05, *p* < 0.01, or *p* < 0.001.

## 5. Conclusions

In summary, our results showed that flavonoid hesperidin attenuates the gingival epithelial barrier dysfunction caused by *P. gingivalis*, and that it exerts a protective effect against *P. gingivalis* adhesion to extracellular matrix proteins. Furthermore, hesperidin provides oxidative stress protection by reducing ROS production by *P. gingivalis*-stimulated oral epithelial cells. This flavonoid attenuated the secretion of pro-inflammatory cytokines and NF-κB activation in macrophages stimulated with *P. gingivalis*. Within the limitations of this study, our promising results suggest that hesperidin may be an excellent candidate for adjuvant therapy for the treatment or prevention of periodontal disease. However, future studies are necessary to assess the exact mechanisms of action of hesperidin on the epithelial barrier and the inflammatory response during periodontal disease. It would be very interesting to undertake studies on the clinical benefits of incorporating hesperidin in oral hygiene products (mouthrinse, toothpaste, gels, and chewing gum) or slow-release devices (inserted in affected periodontal sites).

## Figures and Tables

**Figure 1 ijms-24-10389-f001:**
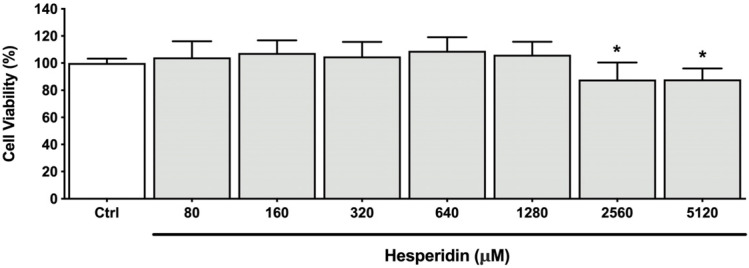
Viability of human oral epithelial cell line B11 after treatment (48 h) with hesperidin. *, significant difference in comparison with the control group (*p* < 0.05).

**Figure 2 ijms-24-10389-f002:**
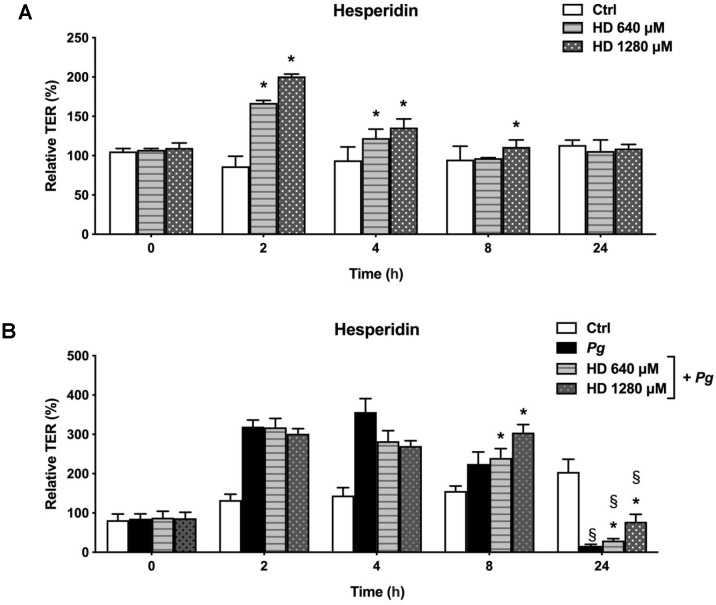
Time- and dose-dependent effects of hesperidin on the oral epithelial barrier model (**A**) and challenged with *P. gingivalis* (**B**) as determined by monitoring the TER. A 100% value was assigned to the TER values recorded at time 0. Results are expressed as the means ± SD of triplicate assays. *, significant increase (*p* < 0.05) compared to control cells (**A**) or *P. gingivalis*-stimulated cells (**B**). §, significant decrease (*p* < 0.05) compared to non-stimulated control cells (**B**).

**Figure 3 ijms-24-10389-f003:**
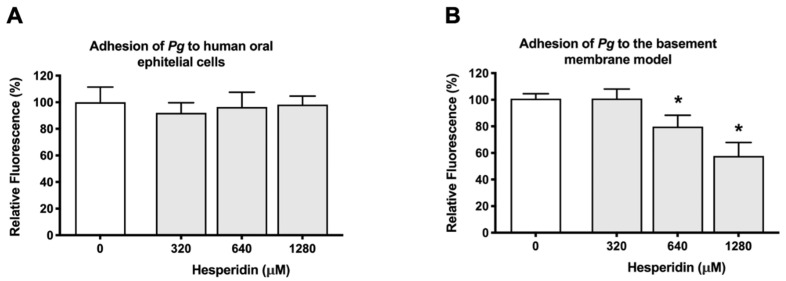
Effect of hesperidin on the adherence of *P. gingivalis* to human oral epithelial cells (**A**) and Cultrex^®^ Basement Membrane Extract (**B**). Results are expressed as the means ± SD to triplicate assays. *, significantly different from the control (*p* < 0.05).

**Figure 4 ijms-24-10389-f004:**
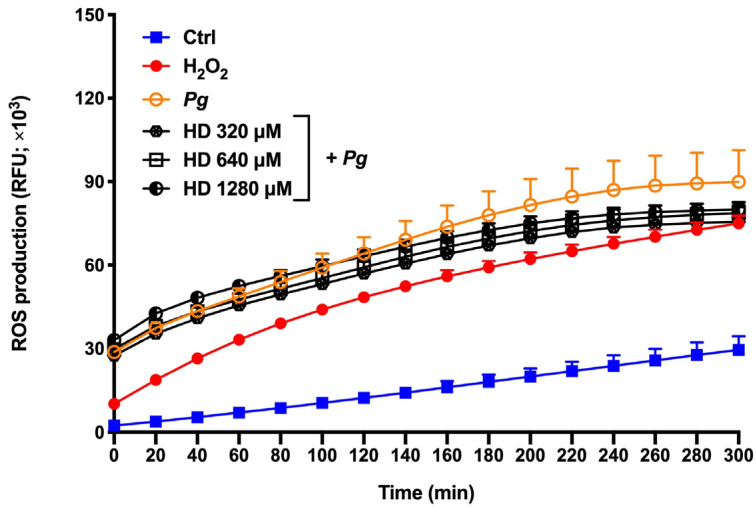
Time- and dose-dependent effects of hesperidin on ROS production by *P. gingivalis*-stimulated epithelial cells. Results are expressed as the means ± SD of three independent experiments. Values are significantly different from *P. gingivalis*-stimulated epithelial cells (*p* < 0.05).

**Figure 5 ijms-24-10389-f005:**
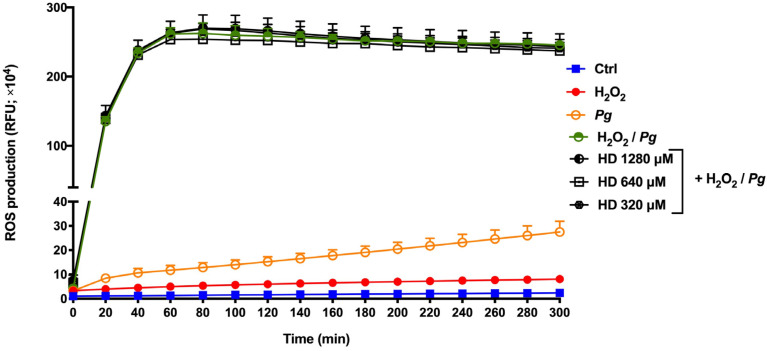
Time- and dose-dependent effects of hesperidin on ROS production by human epithelial cells stimulated synergistically with hydrogen peroxide (H_2_O_2_) and *P. gingivalis*. Results are expressed as the means ± SD of three independent experiments. Values are significantly different between H_2_O_2_ and *P. gingivalis*-stimulated epithelial cells (*p* < 0.05).

**Figure 6 ijms-24-10389-f006:**
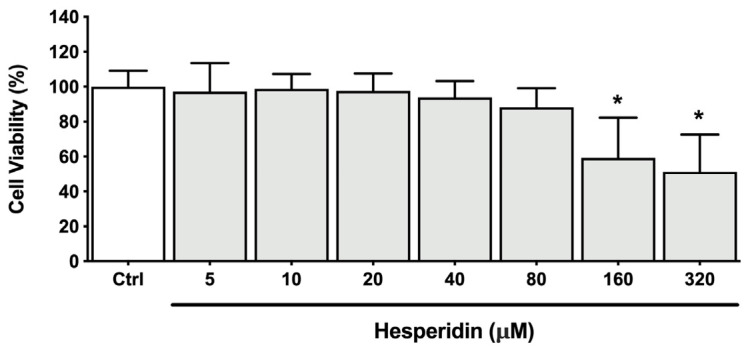
Cell viability of macrophage-like cells after treatment (24 h) with hesperidin. *, significant difference in comparison with the control group (*p* < 0.05).

**Figure 7 ijms-24-10389-f007:**
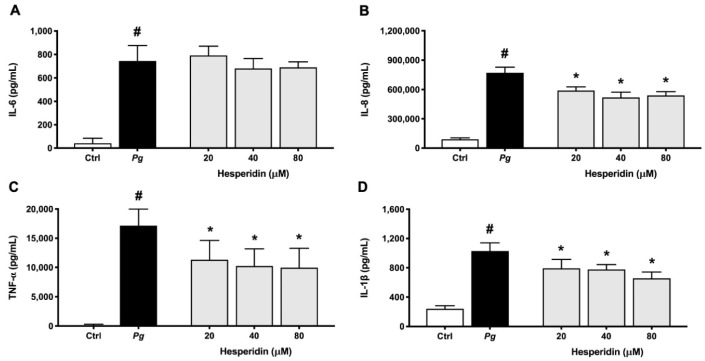
Secretion of IL-6 (**A**), IL-8 (**B**), TNF-α (**C**), and IL-1β (**D**) by macrophage-like cells treated with hesperidin and stimulated with *P. gingivalis* (MOI = 100) for 24 h. Results are expressed as the means ± SD of triplicate assays for three independent experiments. #, significant increase (*p* < 0.001) compared to cells not stimulated with *P. gingivalis*. *, significant decrease (*p* < 0.001) compared to *P. gingivalis*-stimulated cells.

**Figure 8 ijms-24-10389-f008:**
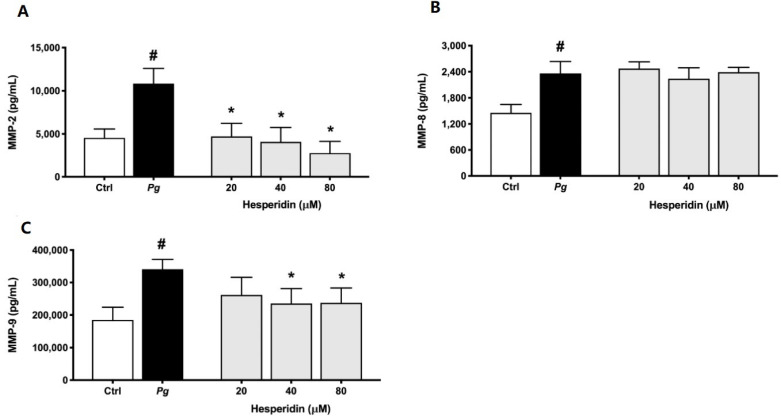
Secretion of MMP-2 (**A**), MMP-8 (**B**), and MMP-9 (**C**) by macrophage-like cells treated with hesperidin and stimulated with *P. gingivalis* (MOI = 100) for 24 h. Results are expressed as the means ± SD of triplicate assays for three independent experiments. #, significant increase (*p* < 0.001) compared to cells not stimulated with *P. gingivalis*. *, significant decrease (*p* < 0.001) compared to *P. gingivalis*-stimulated cells.

**Figure 9 ijms-24-10389-f009:**
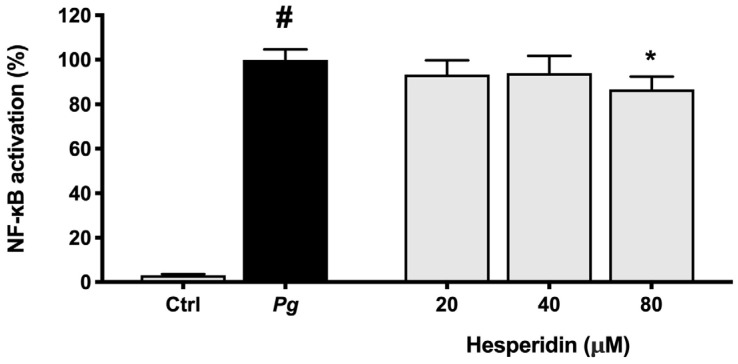
NF-κB activation in the U937-3xκB monocytic cell line induced by *P. gingivalis* (MOI = 100). A value of 100% was assigned to the activation obtained with *P. gingivalis* in the absence of hesperidin. Results are expressed as the means ± SD of triplicate assays for three independent experiments. #, significant increase (*p* < 0.01) compared to cells not stimulated with *P. gingivalis*. *, significant decrease (*p* < 0.01) compared to *P. gingivalis*-stimulated cells.

## Data Availability

The data presented in this study are available upon request from the corresponding author.

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
