# Peer review of "Effect of Hesperidin on Barrier Function and Reactive Oxygen Species Production in an Oral Epithelial Cell Model, and on Secretion of Macrophage-Derived Inflammatory Mediators during Porphyromonas gingivalis Infection"

_ijms, 2023, doi:10.3390/ijms241210389_

Round 1

Reviewer 1 Report

The Article is dedicated to investigation of properties of hesperidine which is being studied as promising agent for therapy and prevention of periodontal diseases. The effect of hesperidin on epithelial barrier function, Porphyromonas gingivalis adherence to a gingival kereatinocites and ROS production by oral epithelial cells, as well as  the secretion of IL-1β, TNF-α, IL-8, MMP-2 and MMP-9 by macrophages challenged with P. gingivalis was demonstrated. Additionally, hesperidin was able to attenuate NF-κB activation in macrophages stimulated with P. gingivalis. These findings elucidate the mechanism of a protective effect of hesperidin on the oral cavity epithelium.

The Abstract presents the goal, the results and methods as well as conclusions briefly.

Introduction is well-written, as well as Discussion, the problem is settled and the current state is analyzed with employing of many references.

The Results are illustrated sufficiently with rather clear Figures.

The Materials and Methods are described in detail and well understandable, it is really fascinating. The Authors have used efficient and adequate methods to obtain the results: a fundamental mechanisms for medicinal use of hesperidin.

The Article may be accepted in present form.

Author Response

General comment: The Article is dedicated to investigation of properties of hesperidine which is being studied as promising agent for therapy and prevention of periodontal diseases. The effect of hesperidin on epithelial barrier function, Porphyromonas gingivalis adherence to a gingival keratinocytes and ROS production by oral epithelial cells, as well as the secretion of IL-1β, TNF-α, IL-8, MMP-2 and MMP-9 by macrophages challenged with P. gingivalis was demonstrated. Additionally, hesperidin was able to attenuate NF-κB activation in macrophages stimulated with P. gingivalis. These findings elucidate the mechanism of a protective effect of hesperidin on the oral cavity epithelium.

The Abstract presents the goal, the results and methods as well as conclusions briefly.

Introduction is well-written, as well as Discussion, the problem is settled and the current state is analyzed with employing of many references.

The Results are illustrated sufficiently with rather clear Figures.

The Materials and Methods are described in detail and well understandable, it is really fascinating. The Authors have used efficient and adequate methods to obtain the results: a fundamental mechanism for medicinal use of hesperidin.

Response: We would like to thank for your valuable comments. We greatly appreciate your attention.

Reviewer 2 Report

This is a well-designed study with important and compelling results. I provide the following suggestions below that would improve this manuscript:

1.     I don’t understand the difference between Figure 3, panels A and B. The x-axis of the panels should be edited to differentiate these two, and this should be described more clearly in the text. Are two different types of hesperidin being used? Or is one a component of hesperidin?

2.     On page 8, line 230, I don’t understand what “protect capacity” is. Protect in what way against what? If this is a particular assay, this should be mentioned.

3.     On page 8, line 263, this sentence is confusing and seems to have incorrect grammar. I think “exacerbate levels” should be “increased levels.”

4.     The authors cannot conclude from these experiments that the effect on inflammatory cytokines is NF-kB-mediated. A different cell line was used, and cytokine inhibition was not measured in that cell line. Even if it were, additional studies would be needed to conclusively prove that the effect is NF-KB-mediated. The authors could instead say on page 9, line 303, that “the effect of hesperidin on pro-inflammatory cytokine expression may be mediated….”

5.     While the effect of hesperidin in vitro is interesting, the most important question is whether these results have clinical significance and if treatment with hesperidin improves outcomes in patients. The results of this study may be statistically significant but still not clinically important. In the discussion or conclusion, the authors should mention that the clinical significance of these findings is unclear and that human studies would be needed to determine whether treatment with hesperidin is beneficial to patients.

This manuscript is written in a clear manner, with the exception of the corrections mentioned in the above review. 

Author Response

General comment: This is a well-designed study with important and compelling results. I provide the following suggestions below that would improve this manuscript:

Point 1: I don’t understand the difference between Figure 3, panels A and B. The x-axis of the panels should be edited to differentiate these two, and this should be described more clearly in the text. Are two different types of hesperidin being used? Or is one a component of hesperidin?

Response 1: Thank you for this comment. As suggested, the figure was modified and some details on this aspect have been added in the revised manuscript (in red).

Point 2: On page 8, line 230, I don’t understand what “protect capacity” is. Protect in what way against what? If this is a particular assay, this should be mentioned.

Response 2: The specific comment was reviewed and the appropriate correction was made in the revised manuscript (in red).

Point 3: On page 8, line 263, this sentence is confusing and seems to have incorrect grammar. I think “exacerbate levels” should be “increased levels.”

Response 3: The specific comment was reviewed and the appropriate correction was made in the revised manuscript (in red).

Point 4: The authors cannot conclude from these experiments that the effect on inflammatory cytokines is NF-kB-mediated. A different cell line was used, and cytokine inhibition was not measured in that cell line. Even if it were, additional studies would be needed to conclusively prove that the effect is NF-KB-mediated. The authors could instead say on page 9, line 303, that “the effect of hesperidin on pro-inflammatory cytokine expression may be mediated….”                                      

Response 4: The specific comment was reviewed and the appropriate correction was made in the revised manuscript (in red).

Point 5: While the effect of hesperidin in vitro is interesting, the most important question is whether these results have clinical significance and if treatment with hesperidin improves outcomes in patients. The results of this study may be statistically significant but still not clinically important. In the discussion or conclusion, the authors should mention that the clinical significance of these findings is unclear and that human studies would be needed to determine whether treatment with hesperidin is beneficial to patients.         

Response 5: We would like to thank the reviewer for this relevant comment. As suggested, some details on this aspect have been added in the revised manuscript (in red).

Reviewer 3 Report

The authors of this manuscript aimed to investigate the effect of Hesperidin on epithelial barrier function, reactive oxygen species production and inflammatory response caused by P. gingivalis. They show that hesperidin protected against gingival epithelial barrier dysfunction and inhibited P. gingivalis mediated ROS production. The manuscript is reasonably well written however, there are many grammatical errors that needs to be addressed.

Ln 41-46. This sentence is too long and therefore makes it harder to understand. Can you make multiple short sentences for clarity.

Ln 88: Replace “withouth” with without

Figure 2A and 2B: I am failing to understand why higher concentrations of Hesperidin were chosen 640 and 1280. Why didn’t the authors go for 80 uM, especially when later on they found concentrations higher than 80 were cytotoxic for macrophages.

Figure 2A: The authors don’t discuss why Hesperidin exposure causes higher TER values? Does Hesperidin cause tighter cell junctions in the epithelial cells?

Figure 2B: why does the TER of the control group increase over time? Shouldn’t that stay same throughout the time course? Also what does it mean when TER increases with P. ging exposure in 2, 4 and 8 hours that is comparable to Hesperidin exposure.

Also in the figures it will be nice to use a bar to show what it is compared to when showing p-value significance

Line 119: it will be nice to start this section by explaining how P. ging were labelled with FITC. i.e FITC binds to the cell surface proteins of P. ging. In general, for results to make sense little background on how it was done always helps in understanding the results.

Figure 3: It will be nice to have a control of no P. ging cells to give an idea of background fluorescence

Please explain the rational of choosing basement membrane model…why would hesperidin have an effect on basement membrane model but not on the epithelial cells.

Subsequent experiments were performed at 80 uM concentrations. So what would be the recommended concentration of Hesperidin to use if this was to be used as a treatment in humans? Seems like 80uM would have no protection effect for epithelial barrier and using concentrations at 1280 would most likely kill all the macrophages in the system.

Ln 343 and 352: fix the way 37oC is written

Ln 363: Can you specify if FITC was conjugated to anything or was an isomer that could bind to proteins?

Ln 362: 50 mM PBS? Do you mean 1x PBS?

The manuscript has several spelling and grammatical errors that need to be corrected. 

Author Response

General comment: The authors of this manuscript aimed to investigate the effect of Hesperidin on epithelial barrier function, reactive oxygen species production and inflammatory response caused by P. gingivalis. They show that hesperidin protected against gingival epithelial barrier dysfunction and inhibited P. gingivalis mediated ROS production. The manuscript is reasonably well written however, there are many grammatical errors that needs to be addressed.

Point 1: Ln 41-46. This sentence is too long and therefore makes it harder to understand. Can you make multiple short sentences for clarity.                                         

Response 1: The specific comment was reviewed and the appropriate correction was made in the revised manuscript (in orange).

Point 2: Ln 88: Replace “withouth” with without 

Response 2: The specific comment was reviewed and the appropriate correction was made in the revised manuscript (in orange).

Point 3: Figure 2A and 2B: I am failing to understand why higher concentrations of Hesperidin were chosen 640 and 1280. Why didn’t the authors go for 80 uM, especially when later on they found concentrations higher than 80 were cytotoxic for macrophages.

Response 3: We decided to use the highest non-cytotoxic concentrations of each cell line. We had the hypothesis that hesperidin could act in a dose-dependent condition. Therefore, we expected that a higher concentration could have a better effect.

Point 4: Figure 2A: The authors don’t discuss why Hesperidin exposure causes higher TER values? Does Hesperidin cause tighter cell junctions in the epithelial cells?        

Response 4: This is an interesting comment. It is indeed possible hesperidin may improve and augment formation of tissue junctions. Some text has been added in the discussion to raise this possibility (in orange).

Point 5: Figure 2B: why does the TER of the control group increase over time? Shouldn’t that stay same throughout the time course? Also what does it mean when TER increases with P. ging exposure in 2, 4 and 8 hours that is comparable to Hesperidin exposure. 

Response 5: Thank you for this comment. TER values increase as cells proliferate and it reaches the 100% confluency. Then, this aspect suggests the intactness of the cell layer. In this experiment, the TER values of the control group reached their maximum after a 24-h treatment of the cells. On the other hand, the increase in TER upon P. gingivalis exposures is due to the number of bacteria adhering to the oral epithelial barrier model increases during different periods, yielding an additional resistance to the TER of the oral epithelial cell layer. Some text has been added in the discussion and results section (in orange).

Point 6: Also in the figures it will be nice to use a bar to show what it is compared to when showing p-value significance

Response 6: All the figures were standardized in their presentation using symbols to highlight the differences between the groups. On the other hand, these details and the p-value are mentioned in detail in the text of the figure.

Point 7: Line 119: it will be nice to start this section by explaining how P. ging were labelled with FITC. i.e FITC binds to the cell surface proteins of P. ging. In general, for results to make sense little background on how it was done always helps in understanding the results.

Response 7: The specific comment was reviewed and the appropriate correction was made in the revised manuscript (in orange).

Point 8: Figure 3: It will be nice to have a control of no P. ging cells to give an idea of background fluorescence

Response 8: The value of the background (control without P. gingivalis) was subtracted from all groups before the statistical analysis. For this reason, this value is not shown as a control group within the figure.

Point 9: Please explain the rational of choosing basement membrane model…why would hesperidin have an effect on basement membrane model but not on the epithelial cells.

Response 9: P. gingivalis cysteine proteinase gingipains are important virulence factors that contribute to the pathogenesis of periodontal disease and degrade extracellular matrix proteins, such as laminin, fibronectin, and type IV collagen. Taking this into account, we decided to use a basement membrane model because it is a solubilized preparation containing several proteins, including laminin, collagen IV, entactin, and heparin sulfate proteoglycans.

Ours results showed that hesperidin has more effect on the P gingivalis adhesion to basement membrane cell. The manufacture indicates that basement membranes provide major barriers to invasion by other cells. Additionally, the proteolytic activity may deteriorate more easily the epithelial cells and consequently the epithelial tissue. Then, this process may facility the penetration to space between epithelial cells and the lamina propria by P. gingivalis.

Some text has been added in the discussion section (in orange).

Point 10: Subsequent experiments were performed at 80 uM concentrations. So what would be the recommended concentration of Hesperidin to use if this was to be used as a treatment in humans? Seems like 80uM would have no protection effect for epithelial barrier and using concentrations at 1280 would most likely kill all the macrophages in the system.

Response 10: It is not possible to recommend a specific concentration of hesperidin to use as a treatment in humans because in vitro and in vivo studies are necessary to elucidate its benefic effect in the treatment of periodontal disease. However, taking into account our results we can suggest that hesperidin have a dose-dependent effect.

Point 11: Ln 343 and 352: fix the way 37oC is written

Response 11: The specific comment was reviewed and the appropriate correction was made in the revised manuscript (in orange).

Point 12: Ln 363: Can you specify if FITC was conjugated to anything or was an isomer that could bind to proteins?

Response 12: We used a FITC isomer.

Point 13: Ln 362: 50 mM PBS? Do you mean 1x PBS?

Response 13: We used in this experimental protocol 50 mM PBS.

Round 2

Reviewer 3 Report

Please review English grammar for the whole paper. For instance, On line 239: we reported interested results...it should be we reported interesting results. These grammatical errors are present throughout the manuscript. All the added extra lines have grammar issues too. 

PBS is a multicomponent buffer. Which component does 50 mM refer to? I think you either used 50 mM phosphate buffer or please define your recipe for PBS. So everyone knows what your PBS is made and which component the 50 mM refers to. 

Please improve on the quality of English. 

Author Response

Thank you for your comments. The English grammar was reviewed and the corrections were made in the revised manuscript (in orange). Also, specific description of buffer was made.

Round 3

Reviewer 3 Report

None

None